# A Review of the Health Benefits of Food Enriched with Kynurenic Acid

**DOI:** 10.3390/nu14194182

**Published:** 2022-10-08

**Authors:** Monika Turska, Piotr Paluszkiewicz, Waldemar A. Turski, Jolanta Parada-Turska

**Affiliations:** 1Department of Molecular Biology, The John Paul II Catholic University of Lublin, 20-708 Lublin, Poland; 2Department of General, Oncological and Metabolic Surgery, Institute of Hematology and Transfusion Medicine, 02-778 Warsaw, Poland; 3Department of Experimental and Clinical Pharmacology, Medical University of Lublin, 20-090 Lublin, Poland; 4Department of Rheumatology and Connective Tissue Diseases, Medical University of Lublin, 20-090 Lublin, Poland

**Keywords:** food, food analysis, food ingredients, infant formula, kynurenic acid, nutrition

## Abstract

Kynurenic acid (KYNA), a metabolite of tryptophan, is an endogenous substance produced intracellularly by various human cells. In addition, KYNA can be synthesized by the gut microbiome and delivered in food. However, its content in food is very low and the total alimentary supply with food accounts for only 1–3% of daily KYNA excretion. The only known exception is chestnut honey, which has a higher KYNA content than other foods by at least two orders of magnitude. KYNA is readily absorbed from the gastrointestinal tract; it is not metabolized and is excreted mainly in urine. It possesses well-defined molecular targets, which allows the study and elucidation of KYNA’s role in various pathological conditions. Following a period of fascination with KYNA’s importance for the central nervous system, research into its role in the peripheral system has been expanding rapidly in recent years, bringing some exciting discoveries. KYNA does not penetrate from the peripheral circulation into the brain; hence, the following review summarizes knowledge on the peripheral consequences of KYNA administration, presents data on KYNA content in food products, in the context of its daily supply in diets, and systematizes the available pharmacokinetic data. Finally, it provides an analysis of the rationale behind enriching foods with KYNA for health-promoting effects.

## 1. Introduction

In 2013, we published a review paper describing the potential role of kynurenic acid (KYNA) in the digestive system. It summarized the presence of KYNA in the lumen of the gastrointestinal tract, and its beneficial health effects in digestive diseases, as well as its presence in food, were summarized. At that time, we pointed out many gaps in the knowledge and understanding of KYNA’s effects outside the central nervous system. However, even then, we assumed that KYNA administration might have some therapeutic potential and, despite many knowledge gaps, we envisaged the relevance of its supplementation [1]. KYNA is a metabolite of tryptophan. It is synthesized endogenously in human and animal bodies and/or absorbed from the digestive system. Although KYNA’s deficiency symptoms have not been described to date, it is reasonable to investigate the benefits of its supplementation. Nowadays, after 10 years of unprecedented progress in the research on KYNA, we can deliberate whether it is legitimate to supplement our diet with KYNA. The current review focuses on the presence of KYNA in food, including estimations of the daily dietary intake of KYNA, its absorption, distribution, and excretion, and it summarizes the scientifically approved health benefits of KYNA administered via the alimentary route, which are not limited to the digestive system.

## 2. Molecular Targets of Kynurenic Acid

Over 40 years of research, several molecular targets on which KYNA acts have been identified (Figure 1). Further research is ongoing and it was recently proposed that KYNA may be a ligand for the hydroxycarboxylic acid receptor 3 (HCAR3) and adrenoceptor alpha-2B (ADRA2B) [2]. However, no direct evidence is available yet to support this hypothesis.

Glutamatergic receptors occur preferentially in the brain, and since KYNA does not cross the blood–brain barrier, their importance is not discussed in the context of alimentary supplementation. Data on this subject can be found in recent review publications [3]. Whether and how KYNA acts on the alpha-7 nicotinic receptor (α7nAChR) is still a matter of scientific debate [4]. Therefore, the effect of KYNA on the aryl hydrocarbon receptor (AhR) and the G protein-coupled receptor 35 (GPR35), which are abundant in peripheral tissues, is of greatest concern. It should, however, be noted that AhR is an intracellular receptor, and so far, no uptake of KYNA has been demonstrated. Thus, it is uncertain whether exogenously administered KYNA can effectively influence these receptors. In this context, studies on KYNA’s peripheral activity have focused mostly on its agonistic action on GPR35. It is worth noting that these receptors are particularly densely represented on both the immune cells and cells of the gastrointestinal tract mucosa [5,6]. 

GPR35 is a rhodopsin-like, 7-transmembrane class A G-protein coupled receptor firstly described in 1998 by O’Dowd et al. [7]. GPR35 gene transcript in humans can be alternatively spliced into variants GPR35a and GPR35b containing 309 and 340 amino-acids, respectively. The differences between these two variants were limited to GPR35 extracellular domain [8]. Six of seventy described single nucleotide polymorphisms in GPR35 gene were indicated as risk variants for inflammatory bowel disease, ankylosing spondylitis, psoriasis, lupus erythematosus, and primary sclerosing cholangitis [9]. 

Activation of GPR35 by endogenous ligands such as KYNA, lysophosphatidic acid (LPA), or mucosal chemokine CXCL17 resulted in internalization of GPR35 and activation of ERK and Rho phosphorylation signaling pathway. A basal activity of GPR35 was connected with Na/K-ATPase pump and induction of Src signaling in epithelial cells responsible for cell proliferation and neovascularization. Moreover, the presence and high expression of GPR35 in intestine together with high concentration of KYNA in its lumen, GPR35 coexpression with both cholecystokinin GPR65 receptor suggest that GPR35 may be a part of the gut-brain signal axis regulating energy balance [10]. 

## 3. Absorption of Kynurenic Acid from the Digestive Tract

### 3.1. Human Studies

The absorption of KYNA from the digestive tract in humans was demonstrated by Kaihara et al. (1956) and by Turska et al. (2019) [11,12]. In the study by Kaihara et al. (1956), synthetic KYNA was suspended in water and ingested by three human subjects. Different doses of KYNA were administered, i.e., 164, 410, and 820 μmole (equivalent to 31, 77.5, and 155 mg, respectively). The urinary excretion of KYNA was measured the day before and on days 1, 2, and 3 after KYNA ingestion. It was found that KYNA administration resulted in its enhanced excretion, especially on the first days after administration [12], which proves that KYNA is absorbed from the digestive tract after its oral ingestion. In the study by Turska et al. (2019), chestnut honey containing KYNA in an amount of 600 μg/g [13] was used. Honey was dissolved in water with a ratio of 1:1 (w:v). Then, 200 mL of the obtained solution was consumed over a 5 min period by both male and female healthy volunteers. All study participants refrained from eating any food and drank only mineral water for 6 h before the study. Venous blood samples were drawn from a peripheral venous catheter at specified times after the administration of the honey beverage: baseline (0), 15, 30, 60, 90, and 120 min. It was found that the mean KYNA concentration in the serum before the administration of chestnut honey was 0.052 ± 0.004 μM and 0.051 ± 0.014 μM in men and women, respectively. Thirty minutes after the ingestion of chestnut honey dissolved in water, the content of KYNA reached its maximum value of 237% and 308% vs. control in men and women, respectively, which proves that KYNA is absorbed from the digestive tract after its oral administration in food. Notably, after 2 h, the level of KYNA was almost back to normal [11].

### 3.2. Animal Studies

The absorption of KYNA from the digestive tract in animals was demonstrated by Kaihara et al. (1956) and by Turska et al. (2018) [12,14]. In the study by Kaihara et al. (1956), KYNA was suspended in water and administered to rats by a stomach tube at the dose of 164 μmole (equivalent to 31 mg). The measurement of the urinary excretion of KYNA was performed on the day before and on days 1, 2, and 3 after KYNA application. It was found that intragastric KYNA administration resulted in its enhanced excretion [12], which proves that KYNA is absorbed from the digestive tract after its alimentary administration in rats. In the study by Turska et al. (2018), labeled 5,7-^3^H-KYNA was dissolved in saline (1 mCi/mL) and administered intragastrically by oral gavage in a volume of 1 mL per 100 g of adult male Swiss mice body weight. Urine, blood, and tissue samples were collected 1, 3, 6, 12, and 24 h after KYNA administration. It was found that labeled KYNA was present in the blood, the spleen, and the liver, as well as in urine [14], proving that KYNA is absorbed from the digestive tract after its intragastric administration in mice.

KYNA administered orally or intragastrically is rapidly absorbed and is present in peripheral blood plasma several minutes after ingestion. The absorption dynamics suggest simple transmission of KYNA in the upper gastrointestinal tract.

## 4. Distribution of Kynurenic Acid

### 4.1. Human Studies

KYNA content in human tissues is shown in Figure 2, presented in Appendix A, and described separately in the respective subsections below.

#### 4.1.1. Serum

The presence of KYNA in human blood serum has been repeatedly reported (Appendix A). The content of KYNA in the serum obtained from adult healthy humans ranges from 0.016 to 0.071 μM and is within the same limits in the serum obtained from children [15,16,17,18,19,20,21,22,23,24,25].

#### 4.1.2. Saliva

The presence of KYNA in human saliva was disclosed by Kuc et al. (2006). In this study, the unstimulated saliva was collected from male adults who refrained from eating, drinking, smoking, and oral hygiene for 2 h before the collection. The presence of KYNA was evidenced in all these saliva samples. The mean concentration of KYNA in the saliva obtained from healthy subjects was 0.0034 μM [26] (Figure 2a).

#### 4.1.3. Gastric Juice

The presence of KYNA in human gastric juice was reported by Paluszkiewicz et al. (2009). Included in this study were 23 adults (15 women and 8 men, mean age: 58 years) who were candidates for nonsteroid anti-inflammatory drugs therapy due to complaints of pain in the course of discopathia and arthrosis. Patients with signs of chronic gastrointestinal disorders or who had proton pump inhibitors were excluded. Upper gastrointestinal endoscopy was performed after 12 h of fasting. Gastric juice was aspirated via the suction channel. The presence of KYNA was determined in all these gastric juice samples. The mean KYNA content was 0.0099 μM. The level of KYNA was neither gender- nor age-dependent [27] (Figure 2a).

#### 4.1.4. Bile

The presence of KYNA in human bile was disclosed by Paluszkiewicz et al. (2009). In this study, bile was obtained from patients with uncomplicated cholecystolithiasis. The group of 18 patients (12 women and 6 men, mean age: 55 years) underwent elective cholecystectomy due to bile stones localized in the gallbladder with uneventful outcomes. The bile samples were obtained extracorporally from the removed gallbladder. The presence of KYNA was determined in all these gallbladder and hepatic bile samples. The mean concentrations of KYNA in the bile obtained from patients with uncomplicated cholecystolithiasis and obstructive jaundice were 0.833 and 0.307 μM, respectively [27] (Figure 2a).

#### 4.1.5. Intestinal Fluid

The presence of KYNA in the human intestinal fluid was reported by Walczak et al. (2011). In this study, 30 ambulatory adult patients scheduled for elective colonoscopy were enrolled. All patients underwent endoscopy after standard bowel cleansing with macrogol. The participants received a single dose of atropine sulphate immediately before the colonoscopy. The colonoscopies were performed under sedation using propofol. The samples (1–2 mL of mucus) were aspirated from the cecum or from the proximal part of the ascending colon. KYNA was detected in all samples obtained from the group of patients without pathological changes in the intestine. Its mean concentration was 0.0822 μM [28] (Figure 2a).

#### 4.1.6. Synovial Fluid

Under physiological conditions, the amount of synovial fluid is too small to be collected for reliable testing. Therefore, the presence of KYNA was detected in the samples of synovial fluid obtained from patients with rheumatoid arthritis, inflammatory spondyloarthropathies, and osteoarthritis, with mean values of 0.0189, 0.0215, and 0.0305 µM, respectively [29].

#### 4.1.7. Sweat

KYNA was detected in sweat samples. Since the density of sweat is very variable, the content of KYNA in sweat was expressed in relation to the sodium content. Two sampling methods were used: an absorbent patch was fixed to the skin for the entire exercise period, or sweat was absorbed at the end of this period by cotton ear tips. In nontrained subjects, the content of KYNA was 14.80 and 8.16 fmol/µg Na, respectively [30].

#### 4.1.8. Cerebrospinal Fluid

The presence of KYNA in human cerebrospinal fluid has been repeatedly reported (Appendix A). The content of KYNA in cerebrospinal fluid obtained from adult healthy humans ranges from 0.0009 to 0.005 μM [16,20,21,31,32,33,34,35,36,37]. The content of KYNA in cerebrospinal fluid obtained from children is within the same limits [38].

#### 4.1.9. Brain

The presence of KYNA in human brain tissue was unequivocally proved by Turski and Schwarcz (1988) [39]. Shortly thereafter, this finding was supported by Moroni et al. (1988), by Connick et al. (1989), and by Swartz et al. (1990) [37,40,41]. In the study by Turski and Schwarcz (1988), human brain tissue was obtained *post mortem* from male subjects (age range: 50–71 years). KYNA was found in all studied brain samples. In this study, the content of KYNA in the human brain varied from 0.14 to 1.58 pmol/mg wet weight tissue (approximately 0.00014–0.00158 μM) in the cerebellum and the caudate nucleus, respectively [39]. Moroni et al. (1988) reported KYNA concentration in human cortex amounting to 0.15 pmol/mg wet weight tissue (approximately 0.00015 μM) [41]. In the study by Swartz et al. (1990), the content of KYNA in human cerebral cortex expressed per proteins amounted to 2.07 pmol/mg protein in the cerebral cortex, and to 3.38 pmol/mg protein in the putamen [37]. A similar range of KYNA content was reported by Baran et al. (2000) in the frontal cortex and the cerebellum, by Schwarcz et al. (2001) in the Brodmann area 9, 10, and 19, and by Baran et al. (2012) in the control frontal cortex and the cerebellum [42,43,44].

#### 4.1.10. Other Organs

There are no data on the content of KYNA in human tissues other than the brain. Our previous review paper presented the content of KYNA in various tissues and organs of rats [1].

#### 4.1.11. Breast Milk

The presence of KYNA in human milk was reported by O’Rourke et al. (2018) and by Milart et al. (2021) [45,46] (Figure 2a). In the paper by O’Rourke et al. (2018), 12 lactating mothers with term babies (>38 weeks) successfully completed the study. Two specimens of hindmilk were collected from all mothers postpartum, on day 7 and day 14 from their homes. The milk samples were manually expressed into sterile polypropylene containers and kept in the fridge until their collection by the researcher. At the laboratory, all samples were stored initially at −20 °C and then at −80 °C until assayed. It was found that the KYNA levels in breast milk were 0.057 and 0.221 μM on day 7 and day 14, respectively [45]. In the study by Milart et al. (2021), breast milk was obtained from 25 healthy breastfeeding women during the first 6 months after labor. The milk samples were collected six times: on day 3 and day 7, at week 2, and in the 1st, 3rd, 4th, 5th, and 6th months after the delivery. The women were instructed on how to collect their breast milk. The samples of human breast hindmilk, after the first breastfeeding of the day, were collected by means of breast pumps, in the amount of 5 mL, to plastic containers and stored in a fridge for no longer than 3 h. Then, placed in human tissue transport boxes, they were delivered to the laboratory, and transferred to glass sterile probes and stored. KYNA was found in all tested samples of human breast milk. The concentration of KYNA increased gradually from 0.021 μM on day 3 to 0.299 μM in the 6th month of breastfeeding [46].

#### 4.1.12. Urine

The presence of KYNA in human urine obtained from adults and children has been repeatedly reported [47,48,49,50,51,52,53,54,55,56,57,58,59,60,61,62] (Figure 2b) (Appendix A). The urine concentration of KYNA ranges between 4.04 and 22.18 μM in adults. Its concentration in children’s urine, as reported by Uberos et al., is 74.07 and 93.12 μM at night- and daytime, respectively [59].

### 4.2. Animal Studies

#### 4.2.1. Distribution of Kynurenic Acid Administered via the Alimentary Route

The detailed distribution pattern of KYNA after its administration via the alimentary route came from an animal study by Turska et al. (2018). In this study, labeled 5,7-^3^H-KYNA was used. Adult male mice were fasted for 12 h before KYNA administration. KYNA dissolved in saline was applied intragastrically by oral gavage in a volume of 1 mL per 100 g of body weight. Urine, blood, and tissue samples were collected 1, 3, 6, 12, and 24 h after KYNA administration and radioactivity was analyzed. It was found that labeled KYNA was present in various parts of the digestive tract: the highest amounts were found in the stomach and the ileum, and lower in the cecum. An unexpectedly high amount of radioactivity was recorded in the intestinal content of the cecum, whereas no accumulation of KYNA was found in internal organs. As can be expected, as early as 1 h after labeled KYNA administration, a very high amount of radioactivity was found in urine [14].

#### 4.2.2. Blood–Brain Barrier

There is a consensus that KYNA does not cross, or only poorly crosses, the blood–brain barrier under normal conditions [63,64,65]. Thus, it is not expected that KYNA administered orally can be distributed into the brain.

#### 4.2.3. Blood–Placental Barrier

KYNA does not cross the placental barrier between the mother and the fetus, as evidenced in the pregnant mice by Goeden et al. [66]. KYNA (10 mg/kg) was administered orally on embryonic day 18. Although a slight but nonsignificant elevation of KYNA was noted, the final conclusion of the study was as follows: “no increase in KYNA levels was observed in the fetal plasma and brain after KYNA itself was given maternally, indicating that peripherally applied KYNA does not cross the placenta”. According to this statement, it is not to be expected that KYNA administered orally can be distributed into the fetus. Independently of the placental barrier function, the concentration of KYNA as high as 1.13 μM in last-trimester amniotic fluid was reported [23]. Amniotic fluid surrounded the fetus and filled the fetal respiratory and digestive lumen during pregnancy. This phenomenon indicated that fetal production of KYNA was initiated independently of the mother’s plasma concentration.

The tissue distribution of KYNA is organ- and system-selective. After peak concentration in blood plasma after ingestion, the highest concentration of KYNA was observed in bile, pancreatic juice, and intestinal lumen, gradually increasing along the intestine. Moreover, the elevation of KYNA administered by digestive route was not recorded in central nervous system and fetus. These findings indicate tight blood–brain and blood–placental barriers for KYNA administered by digestive route in normal conditions. It is thought-provoking that KYNA concentration in milk obtained from breastfeeding mothers gradually increases along breastfeeding time after delivery. This phenomenon suggests specific regulation of KYNA synthesis or excretion in breast tissue; however, data describing such mechanisms were not available to date.

## 5. Metabolism of Kynurenic Acid

### 5.1. Human Studies

Despite some previous research indicating that KYNA might be metabolized to quinaldic acid, it is widely accepted that KYNA is excreted unmetabolized. In 1955, Brown and Price reported the presence of quinaldic acid in amounts of 4.6–6.9 μmol/day in human urine. Moreover, it was found that after the ingestion of tryptophan (39.2 mmol), the content of urine quinaldic acid rose to 45 μmol/day in humans [67]. In 1956, Kaihara et al. showed that the ingestion of KYNA (164–820 μmol) resulted in a 4–8-fold increase in the quinaldic acid content in human urine [12].

### 5.2. Animal Studies

In 1955, Brown and Price reported the presence of quinaldic acid in amounts of 19 µmol in dog urine. Importantly, it was found that after tryptophan (9.8 mmole) ingestion, the content of urine quinaldic acid rose to 96 µmole in dogs. In cats and rats, the content of quinaldic acid was not confirmed [67]. In the study by Kaihara et al. (1956), KYNA was suspended in water and administered to rats by a stomach tube at the dose of 164 μmol (equivalent to 31 mg). It was found that KYNA administration resulted in enhanced excretion of quinaldic acid, which suggests that KYNA is metabolized to quinaldic acid after its alimentary administration [12]. On the other hand, in the study performed by Turski and Schwarcz (1988), labeled ^3^H-KYNA was used. Male adult rats were chronically implanted with a unilateral guide cannula, directed towards the dorsal hippocampus. Seven days after the surgery, intrahippocampal injections of the labeled KYNA were made in unanesthetized rats via an injection cannula. Brain and urine samples were obtained from the same animals. Urine was collected over a period of 30 or 120 min after completion of the intrahippocampal injection. A careful chromatographic analysis failed to reveal the presence of any metabolic product of KYNA, including quinolinic acid, in either the hippocampus or urine [68].

It is generally accepted that KYNA, despite its biologically receptor-related activity, is not metabolized and is described as an end product in tryptophan kynurenine pathway. To date, any metabolic route of KYNA inactivation in studied biological systems has not been identified.

## 6. Excretion of Kynurenic Acid

### 6.1. Human Studies

#### 6.1.1. Urine

The excretion of KYNA in human urine has been repeatedly reported [47,48,49,50,51,52,53,54,55,56,57,58,59,60,61,62] (Appendix A). It can be concluded that the excretion of KYNA in urine ranges from 1.14 to 6.29 mg/day/adult person. According to Uberos et al. (2010), in children, the urinary excretion of KYNA is even higher, ranging from 14.0–17.6 mg/day/person [59]. In another study performed by Molina-Carballo et al. (2021) on children aged 5–14 years, KYNA was found to be 5–7 μg per mg creatinine [60]. After mathematical conversion, the amount of the excreted KYNA can be estimated at the level of 2.64–3.30 mg/day/child. However, it must be considered that these are only two reports and that more detailed studies involving children of different ages are needed before any final conclusions can be drawn.

#### 6.1.2. Feces

The presence of KYNA in the feces of adult humans was evidenced by Dong et al. (2020). In this study, the median and mean content of KYNA was 7.39 and 9.74 nmol/g, respectively. The range was wide, from 0.42 to 29.24 nmol/g [69]. A similar concentration of KYNA, 12.4 nmol/g, was reported in the feces of children by Shestopalov et al. (2020) [25]. Based on these results and an assumption that humans excrete an average of 128 g of fresh feces per person per day [70], it can be calculated that the excretion of KYNA in feces is approximately 0.18–0.23 mg/day/person (Table 1).

#### 6.1.3. Sweat

Saran et al. (2021) reported that the content of KYNA in sweat ranged between 8.16 and 14.80 fmol/μg Na^+^ [30]. Based on these data and an assumption that humans produce 0.5–2 L of sweat/day [71] and excrete Na^+^ in the amount of 0.9 g/L [72], we calculated that the excretion of KYNA in sweat approximated 0.00069–0.00503 mg (Table 1).

#### 6.1.4. Total Daily Excretion of Kynurenic Acid

The loss of KYNA in urine, feces, and sweat was taken into account in the estimation of the total excretion of KYNA (Table 1). The amount of KYNA excreted into the intestinal lumen as gastric juice, bile, and intestinal fluid was not considered separately, because KYNA can be absorbed from the gastrointestinal tract and, even if not all, the unabsorbed KYNA is excreted in the feces. All calculations were performed based on data from publications cited in appropriate sections of the paper. The results of our analyses are presented in Table 1. It can be concluded that the total estimated excretion of KYNA amounts to 1.15–7.0 mg/day/adult person. Most of that, 90–99% is excreted in urine. Feces account for 0.9–10% of KYNA excretion. The amount of KYNA removed in sweat is marginal, less than 0.1% (Table 1).

All available data indicate that KYNA is eliminated in unchanged form by bile, pancreatic juice, and urine. The elimination is observed rapidly after digestive absorption, which confirms that KYNA is only temporarily stored in tissues without any evidence of its accumulation.

## 7. Kynurenic Acid in Food

### 7.1. Human Food

It has been repeatedly shown that KYNA is a natural component of food. Its content in food and food products varies within a wide range of concentrations, from trace amounts up to 2 mg/gram of chestnut honey [73,74,75,76,77,78,79,80] (Figure 3; Appendix A).

#### 7.1.1. Meat

It can be stated that the content of KYNA in meat is low: 0.0014, 0.0031, and 0.0037 μg/g wet weight in beef, pork, and fish, respectively. Even pig liver contains only slightly more KYNA—0.0091 μg/g wet weight. Since KYNA is rapidly excreted from the animal’s body (see Chapter 6), it does not seem possible to easily increase its amount in meat.

#### 7.1.2. Vegetables

Vegetables are a richer source of KYNA (Figure 3; Appendix A). Cauliflower, potato, and broccoli are some of the richest sources, containing KYNA in amounts of 0.0473, 0.1301, and 0.4184 μg/g wet weight, respectively [73]. However, it should be noted that large differences between varieties exist. The comparison of 16 different edible potato varieties grown under similar soil and climatic conditions showed up to 10-fold differences in KYNA content, from 0.04 to 0.65 μg/g wet weight [76]. Similarly, the comparison of yellow- and purple-fleshed potato cultivar Ismena and Provita revealed a threefold difference in KYNA content, 0.226 and 0.683 μg/g wet weight, respectively [81]. The origin of KYNA in plants is poorly understood. Both its synthesis from kynurenine and the absorption of KYNA from the soil were presented [82]. Since KYNA is found in the soil in varying amounts, and in extraordinary large amounts in manure [82], its content in the plant may depend on the site and cultivation method. This may also be the reason for significant differences in KYNA content even in the same type of vegetable. At the same time, it provides an opportunity to increase KYNA content in the plant by appropriate fertilization.

#### 7.1.3. Fruit

The only fruit that has been studied is the apple, which contained 0.0023 μg of KYNA/g wet weight (Appendix A) [73].

#### 7.1.4. Spices and Herbs for Cooking

KYNA content in spices and herbs for cooking was determined in 19 different products (Appendix A). The highest amount of KYNA was found in basil and thyme, 14.08 and 8.87 μg/g wet weight, respectively [77]. The numbers seem to be relatively high in comparison to plants. However, it must be stressed that, generally, commercially marketable spices are dried. Therefore, it is hard to accurately compare the concentration of KYNA in spices and fresh plants.

#### 7.1.5. Honey

Unexpectedly, honey and other bee products contain relatively large amounts of KYNA (Figure 3; Appendix A). An extremely high amount of KYNA was found in chestnut honey, up to 2114.9 μg/g [74]. Interestingly, this applies to chestnut honey obtained from different locations in Europe and Korea [13,74,80]. This is a phenomenon even among other types of honey. KYNA content in popular honey such as sunflower, multiflorous, buckwheat, acacia, and linden honey is 1.73, 0.877, 0.33, 0.181, and 0.179 μg/g, respectively. A high level of KYNA in chestnut honey seems to be related to the high KYNA content in male flowers of the chestnut tree [13].

#### 7.1.6. Dairy

In commercially available dairy products, the content of KYNA is as follows: cow’s milk: 0.017 μg/mL, kefir: 0.2417 μg/mL, yoghurt: 0.2868 μg/mL, white cheese: 0.0766 μg/g, and hard cheese: 0.0084 [42,49] (Figure 3; Appendix A) [73,78].

#### 7.1.7. Fermented Food and Beverages

A relatively high KYNA content was found in fermented food products (Figure 3; Appendix A). It was found in kefir and yoghurt in amounts of up to 0.242 and 0.287 μg/mL, respectively [73,78]. In addition, cocoa powder contains KYNA in the amount of 4.486 μg/g [78]. Interestingly, the alcoholic beverages wine and beer contain KYNA in a broad range of concentrations, up to 0.179 and 0.051 μg/mL, respectively [11,78,79]. These results indicate that the fermenting microorganisms produce KYNA and that this process may significantly increase the KYNA content in food and beverages.

#### 7.1.8. Medicinal Herbs and Supplements

The presence of KYNA has been demonstrated in medicinal herbs and supplements (Appendix A). The highest content expressed on a dry weight basis of the herbs was found in leaves of peppermint, nettle, birch, and horsetail, ranging from 2.27 to 3.82 μg/g dry weight [83]. The intake of KYNA in herbal infusions prepared according to manufacturer’s instructions were found to vary from 1.08 μg/day in the nettle root infusion to 32.5 and 32.6 μg/day in the nettle leaf and St. John’s wort infusion, respectively. Herbal supplements in the form of tablets also contain KYNA. KYNA delivery calculated in a maximum recommended dose of the supplement equals from 0.41 to 30.38 μg/g in chamomile and St. John’s wort tablets, respectively [82].

#### 7.1.9. Baby Food

KYNA was found in human milk [45,46,73] (Appendix A). Interestingly, the content of KYNA in human milk increases more than 14 times during the time of breastfeeding, starting from 0.004 μg/mL on day 3 after labor and reaching a value of 0.057 μg/mL in the 6th month of feeding [46]. KYNA was also found in all 46 artificial baby milk formulas studied. However, in comparison with human milk, in which its content naturally changed over time, the concentration of KYNA in artificial formulas was substantially lower and did not follow its physiological dynamics of changes [46]. In first-food formulas, KYNA content is clearly higher in products containing vegetables (0.0056–0.0148 µg/g) than in meat-based food (0.01 µg/g). In fruit and vegetable juice, its concentration is 0.0019 µg/g [73]. It should be noted that this estimation is based on single measurement results only, which are insufficient to draw definite conclusions.

### 7.2. Animal Food

The presence of KYNA was studied in animal feed for livestock, cats, dogs, and fish. It was shown that KYNA is present in animal feed in varying concentrations. The highest concentration of KYNA was found in feed for livestock, where it varied from 0.198 μg/g fresh weight to 0.414 μg/g fresh weight. Based on the measurement of KYNA content in feed ingredients, the authors concluded that the concentration of KYNA in animal feed was not controlled and deliberately set, but its final content depended on the ingredients used [84].

## 8. Kynurenic Acid Supplementation

### 8.1. Health Effects of Kynurenic Acid Supplementation

Since KYNA is formed endogenously in the body and can also be supplied in food, the question about the relevance of its supplementation is legitimate. No human studies devoted to health effects of exogenously administered KYNA have been conducted to date. An exception is a study in which a solution of chestnut honey was administered to young volunteers and pharmacokinetic parameters of KYNA were determined afterwards. No side effects were reported in this study [11]. Indirectly, the lack of KYNA toxicity can be inferred from studies in which tryptophan was administered to humans. It was found that tryptophan load resulted in an increase in kynurenines, including KYNA, in the blood. Very recently, Sathyasaikumar et al. (2022) reported no serious adverse events and no long-term changes in behavior and health in tryptophan-treated humans who had plasma KYNA levels that increased as much as 145-fold compared to pre-tryptophan values [85].

Further conclusions should be drawn based on the data obtained from studies on rodents. The data search performed showed that there are few publications describing KYNA administration via the alimentary route for the period lasting from 3 days to 2 months (Table 2).

Because KYNA is water-soluble, it was administered in drinking water in most studies. This mode of administration is very convenient because it does not cause stress to the animal. In addition, water was available ad libitum, which allowed KYNA to be taken in the most natural way according to the daily pattern of drinking. The concentrations used ranged from 2.5 to 250 mg/L, approximating a dose of 0.25 to 25 mg/kg of KYNA per 1 kg body weight/day, respectively. Studies have been conducted on both young and adult animals. Generally, no toxic effects of KYNA administration have been reported. When administered to young animals, KYNA did not interfere with their overall growth and development. However, a moderate reduction in the rate of weight gain was observed. This effect was evident in young, but not adult, animals. On the other hand, the reduction of weight gain rate was present in adult rats kept on a high-fat diet (Table 2). These observations allowed for hypothesizing antiobesogenic properties of KYNA during early development [46]. Interestingly, KYNA has been found to stimulate intestinal mucosal growth in young rats and to cause, inter alia, an increase in the intestinal surface area. However, this affects neither the body composition nor bone mineralization and endurance capacity (Table 2). Importantly, the supplementation of KYNA in drinking water to developing rats for 2 months did not impair their brain functions measured in adulthood [92].

In healthy adult mice, the alimentary administration of KYNA did not affect blood hematological parameters. However, experiments performed in vitro on leukocytes and splenocytes obtained from drug supplemented animals revealed that KYNA exerted antioxidant and immunomodulatory effects [87,88]. In spontaneously hypertensive adult rats, the administration of KYNA in drinking water for 3 weeks did not affect the mean arterial pressure, but it moderately reduced heart rate [89] (Table 2).

Another method of the alimentary administration of the drug was utilized by Li et al. (2021) [90]. In this study, KYNA was applied intragastrically to adult mice kept on a high-fat diet, once a day for 8 weeks, in a dose of 5 mg/kg/day. It was found that such a regimen resulted in declined body weight gain and reduced daily energy intake. Moreover, the following serum metabolic parameters were improved: triglyceride, serum high-density lipoprotein cholesterol, and low-density lipoprotein levels. Finally, both the atherosclerosis index and the coronary artery risk index were significantly decreased [90]. Similar metabolic effects of KYNA injected intraperitoneally at a dose of 5 mg/kg/day, once a day for 1 to 4 weeks, to mice on a high-fat diet were described [93]. Very recently, it was reported that KYNA administered intraperitoneally three times decreased the colonization of the intestine by fungi and ameliorated intestinal injury, i.e., inhibited inflammation, promoted the expression of intestinal tight junction proteins, and protected from intestinal barrier damage caused by invasive *Candida albicans* infection in mice [94] (Table 2). Other data indicating multiple health-promoting effects of KYNA come from studies in which the drug was acutely injected intraperitoneally (Appendix A). This route of administration is preferred by researchers because it allows for the accurate dosage and precise determination of the time of action of the substance. Historically, the earliest data relate to the antiulcer effects of KYNA. Glavin and Pinsky, in 1989, showed that KYNA significantly blocked restraint-cold stress ulcers, ethanol ulcers, and basal nonstimulated gastric acid secretion in normal rats. The authors premised both peripheral and cerebral effects of KYNA [95]. Protective effects of injected KYNA on the liver and the pancreas, and in disorders of the lower gastrointestinal tract, were later described (Appendix A). More recently, numerous reports performed on animals on the beneficial effect of KYNA on the conditions commonly referred to as metabolic diseases in humans were published. Antiobesity, cholesterol-lowering, glucose tolerance improvement, and antiatherosclerotic effects were evidenced in appropriate animal models (Appendix A). It is surprising that many of these effects observed after KYNA administration in animals can be therapeutic targets for metabolic syndromes in humans (Figure 4).

Similar disturbances were described as obese-related multimorbidity, when a low-caloric KYNA-enriched functional diet should be considered as supportive care. The effect of KYNA on bone metabolism is also of interest. A recent publication by Shi et al. (2022) revealed that KYNA administered in a relatively low dose of 5 mg/kg/day for 4 weeks alleviated the postmenopausal osteoporosis and highlighted the involvement of the GPR35 receptor, a molecular target of KYNA, in this action [97] (Appendix A).

In addition, it has been shown that KYNA may have beneficial health effects in life-threatening conditions (Appendix A), and this issue requires a separate commentary. Moroni et al. (2012) were the first to communicate that KYNA administered subcutaneously at doses of 500 mg/kg (single injection) or 200 mg/kg three times at 0, 3, and 6 h after LPS dramatically reduced LPS-induced death in mice [98]. The Hungarian group confirmed that KYNA protects against LPS-induced sepsis in subsequent publications, in which a much lower KYNA dose of 30 mg/kg, i.p., was used [99,100]. Most recently, Wang et al. (2022) demonstrated that intraperitoneally administered KYNA (5 mg/kg; three times at days 3, 6, and 9) reduced the mortality of mice infected with *Candida albicans* [94]. Hsieh et al. (2011) reported that KYNA administered intravenously at doses ranging from 30–100 mg/kg attenuated multiorgan dysfunction in rats exposed to heatstroke [101]. Kaszaki et al. (2008) found a profound anti-inflammatory action of KYNA administered in intravenous infusion in experimental colon obstruction in dogs [102]. Similarly, Marciniak, in 2013, demonstrated that the intravenous infusion of KYNA alleviated symptoms of experimental acute pancreatitis in rats [103]. These results deserve special attention because they indicate the feasibility of using KYNA administered as a bolus or an intravenous infusion in life-threatening conditions.

The beneficial wound healing effects of KYNA (Appendix A) after its external administration on the skin and cornea in rabbits are also worth mentioning [104,105]. It is worth noting that a clinical trial with the use of 0.5% KYNA dressing in people with skin scarring has already been successfully completed [106]. Furthermore, KYNA encapsulated in synthetic polymer microspheres implanted in a wound bed in rats was shown to reduce fibrotic tissue formation [107].

Detailed data on the health-promoting effects of KYNA administration in animals regardless of the route of administration are presented in Appendix A [26,87,90,93,94,95,97,98,99,100,101,102,103,104,105,106,108,109,110,111,112,113,114,115,116,117,118,119,120,121,122,123].

Since there are no substantial differences between the effects exerted by KYNA administered alimentary or infused by injection, it can be assumed that its supplementation by the oral route will produce similar effects as those described after infusion.

### 8.2. Clinical Trials

In the ClinicalTrials.gov database, only four registered studies were found in which KYNA was applied to humans. All studies were dedicated to examining the effect of the topical application of KYNA to skin (Appendix A).

### 8.3. Patents

A survey of the PubChem database of patents in which the keyword KYNA appears revealed 23 patent applications dealing with the medical use of the drug (Appendix A). Most of them relate to its oral or injectable administration and only a few assume the topical or local administration of KYNA. The most commonly claimed effect of KYNA is in digestive tract, liver, and pancreatic conditions. KYNA’s effects have also been recommended in cardiovascular pathologies, lipid metabolism disorders, obesity, and kidney dysfunction. In addition, its use in fibrotic diseases, eye diseases, and mental stress is mentioned. In general, this is in line with the data obtained from scientific publications. Some novelty derived from the patent descriptions is KYNA’s effect on skeletal muscles and its proposed use in sarcopenia or hangover control. In addition to the scientific content, it is worth noting that the number of patent applications on the medical use of KYNA has increased substantially since 2016.

## 9. Perspectives

### 9.1. Food as a Dietary Source of Kynurenic Acid

Although the presence of KYNA in food is a topic that has been addressed several times in various publications, the knowledge of its content in food products is still insufficient. It should be underlined that most of the studies come from Eurasia. There are almost no data from other continents and regions where diets are based on other products than in Europe. Thus, it is advisable to continue the search for rich sources of KYNA among various food products from all over the world.

Based on the current knowledge, it can be concluded that KYNA content in food and food products is generally low (Figure 3; Appendix A) and, according to our calculation (Appendix A), its daily intake is very small (Figure 5a) and quite marginal compared to its excretion (Table 1). The calculation of food-related KYNA intake was based on simulation of a dish of cruciferous, yellow, green, and other vegetables mixed equally, assuming that a single serving weighs 80 g according to Bensley et al. (2003) [124]. The KYNA ingested in red meat, poultry, eggs, and other extensively analyzed nutrients is omitted due to trace of their contents. Low intake was established as a consumption of three or fewer servings per week. High intake was established as consumption of more than 10 servings per week.

The calculated mean daily intake of KYNA as a food constituent was estimated to be approximately 0.065 mg in a diet that does not contain honey. The mean daily intake of KYNA derived from food is notably higher when chestnut honey is consumed in significant quantities and may account for 20.81 mg. However, it should be noted that chestnut honey is produced by bees that collect pollen in areas where chestnut trees grow. These regions are limited due to soil and climate requirements as well as cultivation culture [125]. Thus, it can be concluded that, for most of the human population, the daily amount of food-derived KYNA lies well below 0.1 mg. Taking into account that the daily requirement for KYNA in adult humans, calculated on the basis of its daily excretion, varies from 1.2 to 7 mg (Table 1), it can be concluded that, for most people, the food-derived KYNA does not exceed 10% of the daily requirement. In contrast, in people consuming chestnut honey in their diet, the supplied KYNA, approximating the value of 20 mg, almost triples the daily need. The latter estimation points to the safety of the alimentary administration of KYNA in amounts of up to 20 mg/day.

KYNA content in meat is very low, while in vegetables it is considerably higher but there are large differences between individual types of vegetables and their varieties. Moreover, it seems to depend on where and how the plants are grown. For these reasons, it is difficult to make rational dietary recommendations yet. In terms of KYNA content, chestnut honey is a particular exception among food products including other types of honey. On the other hand, almost nothing is known about the KYNA content in fruit. Interestingly, fermented food contains a higher level of KYNA in comparison to unprocessed food. This knowledge can be used to increase KYNA content in the human diet.

It is noteworthy that KYNA is present in breast milk from the beginning of newborn feeding and its content considerably rises over time [46]. The importance of this phenomenon is still far from being understood, but it should be noted that KYNA content in baby food formulas does not parallel its physiological changes.

### 9.2. Chestnut Honey as a Dietary Source of Kynurenic Acid

Summarizing the data collected so far, it is evident that KYNA possesses potential health-promoting relevance (Appendix A). Its presence in food has been repeatedly confirmed, but its amount is relatively small. The exception is chestnut honey, in which the content of this substance can be even 1000 times higher than in other food products (Figure 3; Appendix A). The uniqueness of chestnut honey is reflected by the fact that the consumption of only 10 g of honey provides such an amount of KYNA that corresponds to its average daily excretion. Greater consumption of chestnut honey in the amounts recommended for use for medical purposes [126] can result in the delivery of more than 3000 times KYNA per day than in a standard diet (Figure 5b). This creates a unique opportunity to verify whether the consumption of chestnut honey brings health benefits to humans. In fact, there are scientific reports on the effects of chestnut honey on diseases modeled in laboratory animals [127,128,129,130,131] (Appendix A). Interestingly, similar to KYNA, chestnut honey administered by the alimentary route counteracts the effects of a high-fat diet in mice and also has an antiulcer effect in the alcohol model of the disease in rats [127,128]. Moreover, it was found to enhance the healing of CCl4-induced liver damage in rats [129]. When administered topically, it improves skin wound healing and the healing of corneal alkali burns [130,131]. Other effects described based on in vitro studies and biochemical analyses are summarized in Appendix A [132,133,134,135,136,137,138,139,140,141]. It should be noted that many of the reported effects of chestnut honey parallel those of KYNA. Therefore, it would be interesting to see if a health-promoting effect has been recognized in population-based studies. Unfortunately, we found no relevant studies in the available literature. This seems to be due to the fact that chestnut honey is not identified in publications on the dietary effects on human health. Furthermore, honey is rarely seen as a separate category in such studies. Therefore, we propose to reinvestigate previous data or to conduct such studies including chestnut honey consumption as a separate item in the menu, due to its exceptionally high KYNA content [13]. Such a study may be facilitated by the fact that chestnut honey is not widely consumed worldwide. It is primarily used in areas where chestnut trees grow. In many countries, its consumption is marginal or practically nonexistent. This allows for the matching of appropriate study groups with a high consumption of chestnut honey and a control group that does not consume chestnut honey. A brief analysis of the chestnut cultivation area indicates that it is typical of Mediterranean European countries, but also of some Asian countries. It also appears that many so-called Blue Zones, regions of the world thought to have a higher than usual number of people who live much longer than average, are located in areas where chestnut trees grow.

Chestnut honey is indeed a very rich source of KYNA, but it has some drawbacks and there are significant limitations to its consumption. First of all, honey contains about 50% monosaccharides and, for this reason, its high consumption is not popular nowadays. However, it should be realized that chestnut honey attenuates postprandial glycemic response and, therefore, is classified as a food with a low or medium glycemic index and glycemic load [142,143,144]. Second, due to the presence of bee- and pollen-derived proteins, honey can be allergenic. Young children should not eat honey. Third, the amount of honey produced by bees is limited and may not be enough to cover increased demands. Therefore, other sources of KYNA should be searched for, and at best its safety should be assessed and the use of KYNA as a supplement allowed.

### 9.3. Other Sources of Kynurenic Acid

Recent publications indicate that yeast may be an alternative source of KYNA [79,145,146]. The production of KYNA by yeast can be carried out on an industrial scale and does not require either suitable soil or climatic conditions, as do chestnut cultivation and honey production. This is a major advantage. However, the disadvantages are the relatively low efficiency of the production process and the limited amount of yeast that can be used for feeding. Therefore, further work is needed to improve the process of KYNA production by microorganisms. There is also a lack of data on the amount of KYNA production by the gut microbiome and how this process can be intensified.

## 10. Concluding Remarks

KYNA appears to be a very promising molecule with potential health-promoting importance and beneficial pharmacodynamics. The fact that it is found in very large amounts in chestnut honey, which is consumed only in limited areas, creates the opportunity to trace its effects on the human body without the need for long-term prospective studies. The rationale for such studies appears to be obvious. It is also imperative not to forget to be cautious, and to perform appropriate studies and closely follow information about its potential side effects before KYNA enters widely into food supplementation. Thus, it seems that the question should be when, not if, KYNA will be used as a dietary supplement.

## Figures and Tables

**Figure 1 nutrients-14-04182-f001:**
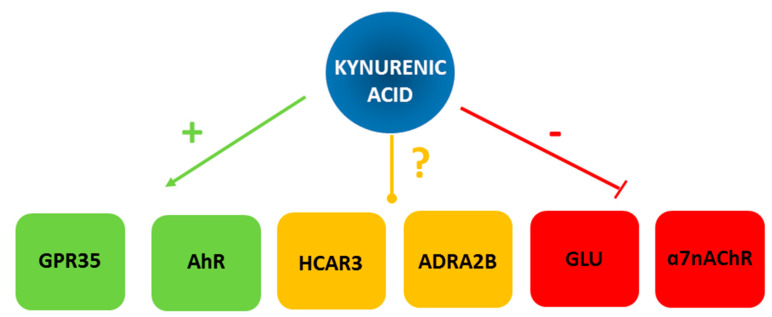
Graphical presentation of molecular targets of kynurenic acid. ADRA2B—adrenoceptor alpha 2B; AhR—aryl hydrocarbon receptor; GPR35—G protein-coupled receptor 35; GLU—glutamate receptor group; HCAR3—hydroxycarboxylic acid receptor 3; α7nAChR—alpha-7 nicotinic acetylcholine receptor; (+)—agonist; (−) antagonist; (?)—putative ligand.

**Figure 2 nutrients-14-04182-f002:**
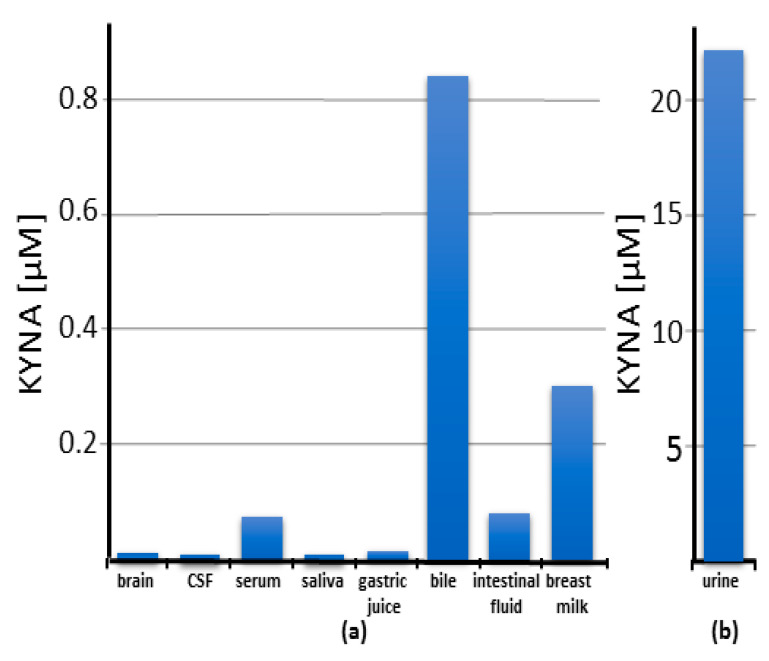
Content of kynurenic acid (KYNA) in human brain and body fluids. Note that the scale on panel (**b**) is 25 times larger than on panel (**a**). CSF—cerebrospinal fluid.

**Figure 3 nutrients-14-04182-f003:**
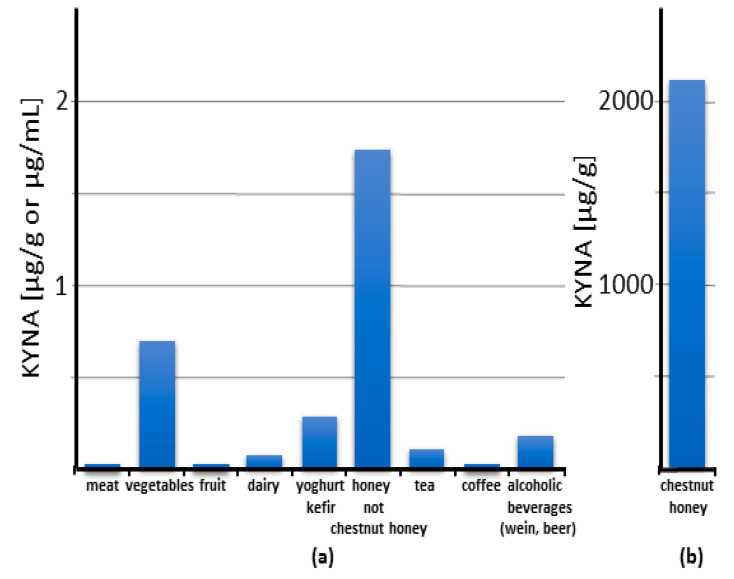
Content of kynurenic acid (KYNA) in food categories: an overview. The columns reflect the value of the highest reported content of KYNA in each category. Note that the scale on panel (**b**) is 1000 times larger than on panel (**a**).

**Figure 4 nutrients-14-04182-f004:**
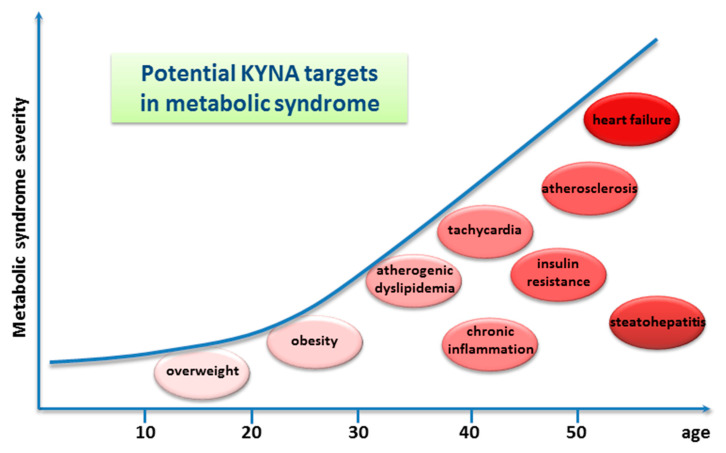
Graphic presentation of potential kynurenic acid (KYNA) targets in metabolic syndrome in humans. The effects of KYNA in specific pathological conditions drawn from animal studies are described in the text and are presented in Appendix A. The graphic is based on [96]; however, only identified KYNA targets are presented.

**Figure 5 nutrients-14-04182-f005:**
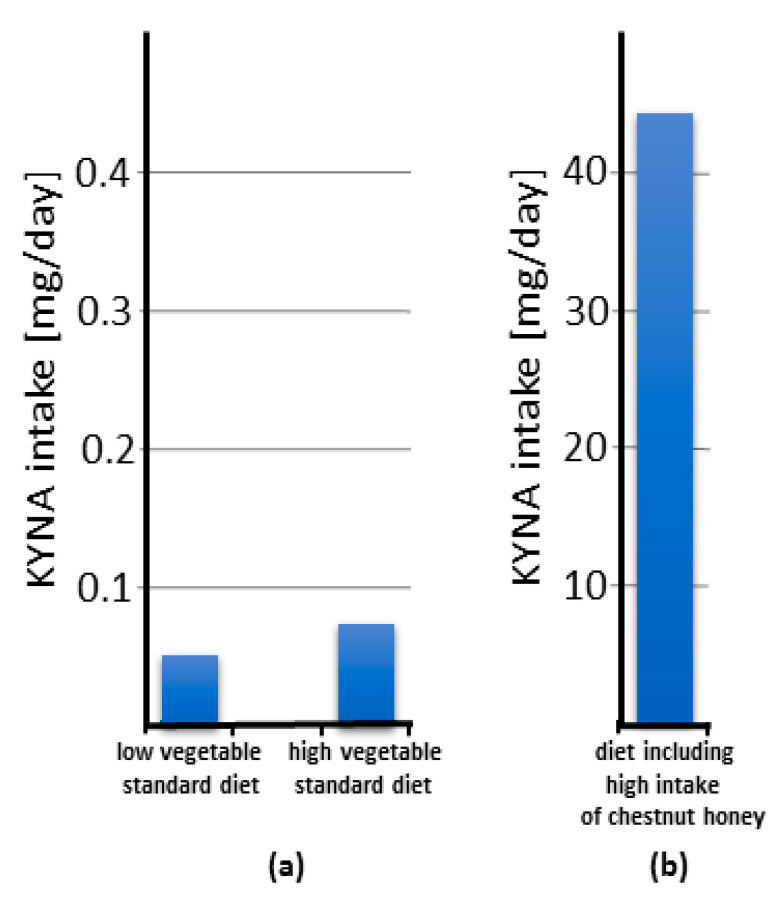
Estimation of the daily intake of kynurenic acid (KYNA) in standard diets and diets rich in chestnut honey. The columns reflect the value of the highest reported content of KYNA (see Appendix A for detail). Note that the scale on panel (**b**) is 100 times larger than on panel (**a**). The daily intake of KYNA in the diet was calculated as follows. The calculation of food-related KYNA intake was based on simulation of a dish of cruciferous, yellow, green and other vegetables mixed equally assuming that a single serving weighs 80 g according to Bensley et al. 2003 [124]. The KYNA ingested in red meat, poultry, eggs and other extensively analyzed nutrients is omitted due to trace of their contents. Low intake was established as consumption of three or fewer servings per week. High intake was established as consumption of more than 10 servings per week.

**Table 1 nutrients-14-04182-t001:** Estimation of daily excretion of kynurenic acid (KYNA) by adult human.

	KYNA Excretion	
	Minimal Level (mg/day)	Maximal Level (mg/day)
Urine	1.14	6.29
Feces	0.010	0.707
Sweat	0.00069	0.00503
Total	1.15	7.00

See body text for references. Calculations of mean KYNA excretion in urine based on data presented in detail in Appendix A.

**Table 2 nutrients-14-04182-t002:** Consequences of kynurenic acid (KYNA) dietary supplementation in rodents.

Species	KYNA Treatment (Dose, Schedule)	Effect/Properties	Reference
Adult animals			
Rats, mice	25 or 250 mg/L in drinking water for 3–21 days	Body weight gain/no effect. Body composition/no effect.	[86]
Mice	2.5, 25, or 250 mg/L in drinking water for 3, 7, 14, 28 days	Activity of peripheral blood leukocytes in vitro/immunomodulation; antioxidant properties.	[87]
Mice	2.5, 25, or 250 mg/L in drinking water for 7–14 days	Hematological parameters/no effect. Splenocytes in vitro/immunomodulatory effect on cytokine production.	[88]
Spontaneously hypertensive rats	25 mg/kg/day in drinking water for 3 weeks	Heart rate/decrease. Mean arterial pressure/no effect.	[89]
Mice	5 mg/kg/day, intragastric; once a day for 8 weeks	High-fat diet induced: Increase of body weight gain/reduction. Increase of daily energy intake/reduction. Increase of serum triglyceride/decrease. Decrease of serum high-density lipoprotein cholesterol/increase. Increase of serum low-density lipoprotein cholesterol/inhibition. Coronary artery risk index/reduction. Atherosclerosis index/reduction. Increase of the ratio of Firmicutes to Bacteroidetes/suppression.	[90]
Young animals			
Rats	25 or 250 mg/L in drinking water; from PND 1 to PND 60	Body weight gain/attenuation. Skeleton development/no effect on bone densitometry and biomechanical endurance.	[91]
Rats	250 mg/L in drinking water; from PND 1 to PND 21	Body weight gain/attenuation. Bone mineral density/no effect. Morphological changes in jejunum/increase in both intestinal surface absorption area and mucosa thickness.	[46]
Rats	25 mg/L in drinking water; from PND 21 until 9th week of life	Open field and locomotor activity tests/no effect. Memory tests/no effect. Depressive and anxiety tests/no effect. Kynurenic acid content in blood and brain/no effect. Kynurenine aminotransferases activity in brain tissue/no effect.	[92]

## Data Availability

Not applicable.

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
