# Peer review of "A Review of the Health Benefits of Food Enriched with Kynurenic Acid"

_nutrients, 2022, doi:10.3390/nu14194182_

Round 1

Reviewer 1 Report

Turska et al. aimed to review the absorption, distribution, excretion, benefits, and food sources of KYNA in human and animal studies. This will provide important insights for more deep research studies and general readers.

Q1: More elaboration on sub-title 2 (Molecular targets of kynurenic acid)

Scientific findings are always under debated and this how they make progress. Instead of simply claiming scientific debate or citing the related review, the authors of this manuscript should elaborate appropriate mechanisms of action a little bit more, such as Line 51, Line 62, Line 64. Particularly, since GPR 35 is the most studied receptor that KYNA serves as an agonist, the detailed mechanism of action should be elaborated well in this subsection. If possible, cartoon signaling graphs can be added to Figure 1.

Q2: Non-essential details of cited research can be eliminated to simplify the main points, such as Line 103-104 (vendor of chemicals). The authors should check the entire manuscript to avoid such cases.

Q3: Missing summary for each subsection.

An independent summary paragraph should be provided at the end of each subsection to highlight the commonality from human and animal studies, such as “3. Absorption of kynurenic acid from the digestive tract”, “4. Distribution of kynurenic acid” , “5. Metabolism of kynurenic acid”, etc..

Q4: Outdated references for subsection “5. Metabolism of kynurenic acid”

The referred studies were between 1955 and 1988, which were a bit outdated. More recent studies regarding this topic need to be discussed and incorporated here.

Q5: More elaboration wanted on “8.1. Health effects of kynurenic acid supplementation”.

The authors did not discuss the benefits of taking KYNA in these two referred human studies besides no severe adverse effects. Considering the goal of this review and the interest of the readers, human studies were much less discussed compared to animal studies on KYNA benefits. I would suggest the authors discuss more human studies.

Q6: The word of “adult” can be removed under the “Species” of Table 2, since it’s under the sub-title “adult animals”. Similarly, the word of “young” can be removed since it’s under sub-title “young animals”.

Q7: Figure 4 is miss leading.

The figure 4 was depicted purely based on animal studies described in the context. If the authors intend to identify KYNA targets in metabolic syndrome, they should mention clearly that these conclusions were drawn from animal studies, thus just showing an indication in MS. Otherwise, the authors should consider removing figure 4 from the manuscript.

Q8: I would suggest removing the word “expected” from the title. The corrected title should be “A review on the health benefits of food enriched with kynurenic acid”.

Author Response

Response to comments of Reviewer I

Q1: More elaboration on sub-title 2 (Molecular targets of kynurenic acid)

Scientific findings are always under debated and this how they make progress. Instead of simply claiming scientific debate or citing the related review, the authors of this manuscript should elaborate appropriate mechanisms of action a little bit more, such as Line 51, Line 62, Line 64. Particularly, since GPR 35 is the most studied receptor that KYNA serves as an agonist, the detailed mechanism of action should be elaborated well in this subsection. If possible, cartoon signaling graphs can be added to Figure 1.

The cumulated data about the role of GPR35 in physiological and pathological condition can be a subject of separate review. However, comments from both Reviewers indicate the need to expand the information on the GPR35 receptor. According to these suggestion following sentences were introduced into the text:

“GPR35 is a rhodopsin-like, 7-transmembrane class A G-protein coupled receptor firstly described in 1998 by O’Dowd et al. [O'Dowd BF, Nguyen T, Marchese A, Cheng R, Lynch KR, Heng HH, et al. Discovery of Three Novel G-Protein-Coupled Receptor Genes. Genomics (1998) 47:310–3. doi: 10.1006/geno.1998.5095]. GPR35 gene transcript in humans can be alternatively spliced into variants GPR35a and GPR35b containing 309 and 340 amino-acids, respectively. The differences between these two variants were limited to GPR35 extracellular domain [Okumura S, Baba H, Kumada T, Nanmoku K, Nakajima H, Nakane Y, et al. Cloning of a G-Protein-Coupled Receptor That Shows an Activity to Transform NIH3T3 Cells and is Expressed in Gastric Cancer Cells. Cancer Sci (2004) 95:131–5. doi: 10.1111/j.1349-7006.2004.tb03193.x]. Six of seventy described single nucleotide polymorphisms in GPR35 gene were indicated as risk variants for inflammatory bowel disease, ankylosing spondylitis, psoriasis, lupus erythematosus and primary sclerosing cholangitis [Kaya B, Melhem H and Niess JH (2021) GPR35 in Intestinal Diseases: From Risk Gene to Function. Front. Immunol. 12:717392. doi: 10.3389/fimmu.2021.717392].

Activation of GPR35 by endogenous ligands like KYNA, lysophosphatidic acid (LPA) or mucosal chemokine CXCL17 resulted in internalization of GPR35 and activation of ERK and Rho phosphorylation signaling pathway. A basal activity of GPR35 was connected with Na/K-ATPase pump and induction of Src signaling in epithelial cells responsible for cell proliferation and neovascularization. Moreover, GPR35 and their expression along intestine with high concentration of KYNA is responsible for energy homeostasis by cholecystokinin, release of pancreatic enzyme, increase glucose tolerance and co-expression with proton-sensitive receptor GPR65. The regulation of total cellular energy expenditure, food intake and metabolism suggest the role of GPR35 in gut-brain signal axis [Quon T. et al. ACS Pharmacol Trans Sci, 2020, 3, 801-12].”

Q2: Non-essential details of cited research can be eliminated to simplify the main points, such as Line 103-104 (vendor of chemicals). The authors should check the entire manuscript to avoid such cases.

The non-essential details of cited research were eliminated to clarifying text. The corrections were performed in Lines: 102, 105, 206

Q3: Missing summary for each subsection.

An independent summary paragraph should be provided at the end of each subsection to highlight the commonality from human and animal studies, such as “3. Absorption of kynurenic acid from the digestive tract”, “4. Distribution of kynurenic acid” , “5. Metabolism of kynurenic acid”, etc..

Suggested summary sentences were inserted into each subsection according to Reviewer suggestions. The sentences were introduced in the end of all subsections as follows:

    1. Absorption of …

“KYNA administered orally or intragastrically is rapidly absorbed and is present in peripheral blood plasma several minutes after ingestion. The absorption dynamics suggest simple transmission of KYNA in upper gastrointestinal tract.”

    1. Distribution of …

“The tissue distribution of KYNA is organ and system-selective. After peak concentration in blood plasma after ingestion the highest concentration of KYNA was observed in bile, pancreatic juice and intestinal lumen, gradually increasing along intestine. Moreover, the elevation of KYNA administered by digestive route was not recorded in central nervous system and fetus. These finding indicate sufficient normal blood-brain and placental barriers for KYNA administered by digestive route. It is thought-provoking that KYNA concentration in milk obtained from breast-feeding mothers’ gradually increases along breast-feeding time after delivery. This phenomenon suggests specific regulation of KYNA synthesis or excretion in breast tissue, however data describing such mechanisms was not available to date.”

    1. Metabolism of…

“It is generally accepted that KYNA, despite of their biologically receptor-related activity, is not metabolized and is described as end-product in tryptophan kynurenine pathway. To date, any metabolic route of KYNA inactivation in studied biological systems has not been identified.”

    1. Excretion of…

“All available data indicates that KYNA is eliminated in unchanged form by bile, pancreatic juice and urine. The elimination is observed rapidly after digestive absorption confirms that KYNA is only temporarily stored in tissues without any evidences on its accumulation.”

Q4: Outdated references for subsection “5. Metabolism of kynurenic acid”

The referred studies were between 1955 and 1988, which were a bit outdated. More recent studies regarding this topic need to be discussed and incorporated here.

It is generally accepted that KYNA is end product of tryptophan catabolic pathway known as kynurenine route. Any data not indicate, that KYNA administered by digestive route is metabolized to any related substance. The data obtained from studies using radio-labeled KYNA shows, that absorption, tissues storing and excretions processes confirmed the presence of unchanged KYNA with retention of tritium incorporated into KYNA particle. These data were shortly indicated as summarized sentences in the end of subsections 5 and 6.

See also answer to Q5.

Q5: More elaboration wanted on “8.1. Health effects of kynurenic acid supplementation”.

The authors did not discuss the benefits of taking KYNA in these two referred human studies besides no severe adverse effects. Considering the goal of this review and the interest of the readers, human studies were much less discussed compared to animal studies on KYNA benefits. I would suggest the authors discuss more human studies.

In the manuscript two publications that refer to the study of KYNA's effects on humans, namely Kaihara et al., 1956, and Turska et al., 2019 were cited. These are the only publications we found in the available databases. In section 8.1 we wrote: “No human studies devoted to health effects of exogenously administered KYNA have been conducted to date.” The reason we wrote this review paper is actually due to the lack of research on KYNA's health effects on humans. We hope that the compilation of data derived from animal studies will prompt researchers to undertake appropriate human studies.

Q6: The word of “adult” can be removed under the “Species” of Table 2, since it’s under the sub-title “adult animals”. Similarly, the word of “young” can be removed since it’s under sub-title “young animals”.

Table 2 was corrected.

Q7: Figure 4 is miss leading.

The figure 4 was depicted purely based on animal studies described in the context. If the authors intend to identify KYNA targets in metabolic syndrome, they should mention clearly that these conclusions were drawn from animal studies, thus just showing an indication in MS. Otherwise, the authors should consider removing figure 4 from the manuscript.

It is true, the figure is based on the results of a study that was conducted on animals. Therefore, the legend states that these are “potential KYNA targets”. To further emphasize that Figure 4 is based on research conducted on animal models the following changes were made:

    • in the heading shown in the figure, "identified" was changed to "potential";
    • in the legend to Figure 4, the following phrases were added: “in humans” and “drawn from animal studies. The legend reads: “Graphic presentation of potential kynurenic acid (KYNA) targets in metabolic syndrome in humans. The effects of KYNA in specific pathological conditions drawn from animal studies are described in the text and presented in Table S7.”
    • in body text, the following phrases were added: “performed on animals”, “in animals” and in humans. The relevant part of the text reads as follows: “More recently, numerous reports performed on animals on the beneficial effect of KYNA on the conditions commonly referred to as metabolic diseases in humans were published. Antiobesity, cholesterol-lowering, glucose tolerance improvement and antiatherosclerotic effects were evidenced in appropriate animal models (Table S7). It is surprising that many of these effects observed after KYNA administration in animals can be therapeutic targets for metabolic syndromes in humans (Figure 4).”

Q8: I would suggest removing the word “expected” from the title. The corrected title should be “A review on the health benefits of food enriched with kynurenic acid”.

Thank you very much for your brilliant suggestion. The title of article was changed.

Reviewer 2 Report

In this review, Murska et al. summarized the health benefits of kynurenic acid in humans and discussed the role of kynurenic acid in various pathological conditions. Generally, the manuscript is well thought out and well organized. However, I would like to point out some minor corrections.

1. In this review, the authors focused mainly on KYNA’s agonistic activity of GPR35. But there is no discussion on how KYNA modulates the GPR35 activity. I think the authors need to discuss KYNA’s metabolism and its action on GPR35.

2. Line 122: According to table S1, is the upper limit 0.071 instead of 0.06?

3. Line 551: Shouldn’t it be Table S4 instead of S7?

4. Table 1: The estimation of daily excretion of KYNA in feces in the table doesn’t match with the numbers in the text in paragraph 6.1.2.

5. Table 1: These are three independent studies, and it is mathematically incorrect to add the numbers to calculate the percentages. There is no meaning of % columns in the context.

6. The writing should be considerably improved.

a. several typographical/grammar errors need correcting:

·      Figure 1 shows a space in GPR 35 and in É‘7 nAChR. But in the text (lines 63, 65), no space was used for GPR35, É‘7nAChR. I suggest the authors to edit figure 1 with no space in the receptor names.

·      Correct usage of articles: For example, line 109: in ‘the’ blood.

·      Line 151: in ‘the’ human….

·      Line 190: ‘A’ similar range….

·      Line 544: Eurasia (no ‘the’)

·      Line 618: with ‘a’ high

b. Line 212: instructed ‘on’ how to….

    Line 291: ‘from’ 1.14-6.29

    Line 297: remove ‘that’

    Line 306: feces (not faces)

    Line 325: less than 1 per (mille?)

    Line 309: detail (no plural)
    Line 393: instructions ‘were’ found

    Line 455-458: Rewrite the sentence for clarity.

    Line 488: syndromes (plural noun)

    Line 501: 500 mg/kg (is it per day?)

    Line 505: reduced the mortality of mice (by what percent?)

    Line 586: considerably ‘rises’

    Line 588: formulas ‘do’ not

    Line 599: benefits ‘to humans’

    Line 603: Moreover, was ‘it’?

    Line 626: high ‘consumption’ is…

    Line 628: as a food ‘with’ …

    Line 661: investigate ‘the’ kynurenic acid…

    Line 666: animal(s)

    Lines 342- 344: Provide the correct citation.

Author Response

Response to comments of Reviewer II

  1. In this review, the authors focused mainly on KYNA’s agonistic activity of GPR35. But there is no discussion on how KYNA modulates the GPR35 activity. I think the authors need to discuss KYNA’s metabolism and its action on GPR35.

The cumulated data about the role of GPR35 in physiological and pathological condition can be a subject of separate review. However, since comments from both Reviewers indicate the need to expand the information on the GPR35 receptor the following sentences were introduced into the text.

“GPR35 is a rhodopsin-like, 7-transmembrane class A G-protein coupled receptor firstly described in 1998 by O’Dowd et al. [O'Dowd BF, Nguyen T, Marchese A, Cheng R, Lynch KR, Heng HH, et al. Discovery of Three Novel G-Protein-Coupled Receptor Genes. Genomics (1998) 47:310–3. doi: 10.1006/geno.1998.5095]. GPR35 gene transcript in humans can be alternatively spliced into variants GPR35a and GPR35b containing 309 and 340 amino-acids, respectively. The differences between these two variants were limited to GPR35 extracellular domain [Okumura S, Baba H, Kumada T, Nanmoku K, Nakajima H, Nakane Y, et al. Cloning of a G-Protein-Coupled Receptor That Shows an Activity to Transform NIH3T3 Cells and is Expressed in Gastric Cancer Cells. Cancer Sci (2004) 95:131–5. doi: 10.1111/j.1349-7006.2004.tb03193.x]. Six of seventy described single nucleotide polymorphisms in GPR35 gene were indicated as risk variants for inflammatory bowel disease, ankylosing spondylitis, psoriasis, lupus erythematosus and primary sclerosing cholangitis [Kaya B, Melhem H and Niess JH (2021) GPR35 in Intestinal Diseases: From Risk Gene to Function. Front. Immunol. 12:717392. doi: 10.3389/fimmu.2021.717392]

Activation of GPR35 by endogenous ligands like KYNA, lysophosphatidic acid (LPA) or mucosal chemokine CXCL17 resulted in internalization of GPR35 and activation of ERK and Rho phosphorylation signaling pathway. A basal activity of GPR35 was connected with Na/K-ATPase pump and induction of Src signaling in epithelial cells responsible for cell proliferation and neovascularization. Moreover, GPR35 and their expression along intestine with high concentration of KYNA is responsible for energy homeostasis by cholecystokinin, release of pancreatic enzyme, increase glucose tolerance and co-expression with proton-sensitive receptor GPR65. The regulation of total cellular energy expenditure, food intake and metabolism suggest the role of GPR35 in gut-brain signal axis [Quon T. et al. ACS Pharmacol Trans Sci, 2020, 3, 801-12].”

  1. Line 122: According to table S1, is the upper limit 0.071 instead of 0.06?

corrected

  1. Line 551: Shouldn’t it be Table S4 instead of S7?

corrected

  1. Table 1: The estimation of daily excretion of KYNA in feces in the table doesn’t match with the numbers in the text in paragraph 6.1.2.

The data presented in the Table 1 were recalculated and corrected in the text (table).

  1. Table 1: These are three independent studies, and it is mathematically incorrect to add the numbers to calculate the percentages. There is no meaning of % columns in the context.

The columns presented percent of excreted KYNA were deleted from Table 1.

  1. The writing should be considerably improved.

a. several typographical/grammar errors need correcting:

  • Figure 1 shows a space in GPR 35 and in É‘7 nAChR. But in the text (lines 63, 65), no space was used for GPR35, É‘7nAChR. I suggest the authors to edit figure 1 with no space in the receptor names.

corrected

  • Correct usage of articles: For example, line 109: in ‘the’ blood.

corrected

  • Line 151: in ‘the’ human….

corrected

  • Line 190: ‘A’ similar range….

corrected

  • Line 544: Eurasia (no ‘the’)

corrected

  • Line 618: with ‘a’ high

corrected

  1. Line 212: instructed ‘on’ how to….

corrected

    Line 291: ‘from’ 1.14-6.29

corrected

Line 297: remove ‘that’

corrected

Line 306: feces (not faces)

corrected

Line 325: less than 1 per (mille?)

corrected

    Line 309: detail (no plural)

corrected

    Line 393: instructions ‘were’ found

corrected

    Line 455-458: Rewrite the sentence for clarity.

The sentence was corrected as follows: “In healthy adult mice, the alimentary administration of KYNA did not affect blood hematological parameters. However, experiments performed in vitro on leukocytes and splenocytes obtained from drug supplemented animals revealed that KYNA exerted antioxidant and immunomodulatory effects”.

    Line 488: syndromes (plural noun)

corrected

    Line 501: 500 mg/kg (is it per day?)

The sentence was corrected as follows: “Moroni et al, 2012, were the first to communicate that KYNA administered subcutaneously at doses of 500 mg/kg (single injection) or 200 mg/kg three times at 0; 3 and 6 h after LPS dramatically reduced LPS-induced death in mice.”

    Line 505: reduced the mortality of mice (by what percent?)

It was not specified by Authors. There is the following statement in the original publication: “Mice infected with C. albicans experience higher mortality than mice treated with KynA [P = 0.01, HR 2.22 (95% CI: 1.06–4.64)]”.

    Line 586: considerably ‘rises’

corrected

    Line 588: formulas ‘do’ not

corrected

    Line 599: benefits ‘to humans’

corrected

    Line 603: Moreover, was ‘it’?

corrected

    Line 626: high ‘consumption’ is…

corrected

    Line 628: as a food ‘with’ …

corrected

    Line 661: investigate ‘the’ kynurenic acid…

corrected

    Line 666: animal(s)

corrected

    Lines 342- 344: Provide the correct citation.

Citation was added.